# Social selectivity and social motivation in voles

Annaliese K Beery[1,2,3]*, Sarah A Lopez[2], Katrina L Blandino[2], Nicole S Lee[3], Natalie S Bourdon[2]

[1]Department of Integrative Biology, University of California Berkeley, Berkeley, United States; [2]Program in Neuroscience, Departments of Psychology and Biology, Smith College, Northampton, United States; [3]Neuroscience and Behavior Graduate Program, University of Massachusetts, Amherst, MA, United States

**Abstract** Selective relationships are fundamental to humans and many other animals, but relationships between mates, family members, or peers may be mediated differently. We examined connections between social reward and social selectivity, aggression, and oxytocin receptor signaling pathways in rodents that naturally form enduring, selective relationships with mates and peers (monogamous prairie voles) or peers (group-living meadow voles). Female prairie and meadow voles worked harder to access familiar versus unfamiliar individuals, regardless of sex, and huddled extensively with familiar subjects. Male prairie voles displayed strongly selective huddling preferences for familiar animals, but only worked harder to repeatedly access females versus males, with no difference in effort by familiarity. This reveals a striking sex difference in pathways underlying social monogamy and demonstrates a fundamental disconnect between motivation and social selectivity in males—a distinction not detected by the partner preference test. Meadow voles exhibited social preferences but low social motivation, consistent with tolerance rather than reward supporting social groups in this species. Natural variation in oxytocin receptor binding predicted individual variation in prosocial and aggressive behaviors. These results provide a basis for understanding species, sex, and individual differences in the mechanisms underlying the role of social reward in social preference.

**\*For correspondence:**
abeery@berkeley.edu

**Competing interest:** The authors declare that no competing interests exist.

## Introduction

The brain regions and neurochemicals involved in social behaviors show remarkable conservation across species (*O'Connell and Hofmann, 2011*). At the same time, social behavior is not a unified construct, with different species exhibiting distinct social structures and behavioral repertoires. The formation of selective social relationships is a particular hallmark of both human and prairie vole societies. Such relationships are difficult to study in traditional lab rodents because mice, rats, and other rodents typically do not form preferences for known peers or mates (*Triana-Del Rio et al., 2015*; *Schweinfurth et al., 2017*; *Beery et al., 2018*; *Cymerblit-Sabba et al., 2020*; *Insel et al., 2020*; *Beery and Shambaugh, 2021*). In species that form specific relationships, selectivity may be based on reward and prosocial motivation toward specific individuals, or on avoidance (fear, aggression) of unfamiliar individuals. The role of social motivation and tolerance may also differ by familiarity, sex, and type of relationship (e.g. same-sex peer versus opposite-sex mate). Voles provide an opportunity to probe the role of selectivity and social reward across relationship types and social organization.

The reinforcing properties of social interaction have been demonstrated in a variety of rodent species and contexts, often through conditioned place preference for a socially associated environmental cue (e.g. *Panksepp and Lahvis, 2007*; *Dölen et al., 2013*; *Goodwin et al., 2019*). Operant conditioning for access to a social stimulus has been used to more directly measure motivation for

**eLife digest** What factors drive the formation of social relationships can vary greatly in animals. While some individuals may be motivated to find social partners, others may just tolerate being around others. A desire to avoid strangers may also lead an individual to seek out acquaintances or friends. Sometimes a mix of these factors shape social behavior.

Studying motivation for social relationships in the laboratory is tricky. Traditional laboratory animals like mice and rats do not bond with specific peers or mates. But small burrowing rodents called voles are a more relationship-oriented alternative to mice and rats. Prairie voles form selective and enduring preferences for both their mates and familiar same-sex peers. Meadow voles on the other hand, live alone much of the year but move in with other animals over the winter.

Beery et al. show that social motivation in voles varies by relationship type, species and sex. In the experiments, voles were first trained to press a lever to get a food reward. Then, the food reward was swapped with access to familiar or unfamiliar voles. Female prairie voles strived to be with animals they knew rather than to be with strangers, while male prairie voles tried hard to access any female. In contrast, meadow voles did not overly exert themselves to access other animals.

Beery et al. then measured oxytocin receptor levels in the brains of prairie voles. Prairie voles that had more receptors for oxytocin in part of their brain known as the nucleus accumbens worked harder to access their familiar partner. But individuals with more oxytocin receptors in the bed nucleus of the stria terminalis were more likely to attack an unfamiliar animal.

The meadow voles' behavior suggests that they are more motivated by tolerance of familiar animals, while the female prairie voles may find it rewarding to be with animals they have bonded with. These differences may help explain why these two species of vole have evolved different social behaviors. The experiments also suggest that oxytocin – which is linked with maternal behavior – plays an important role in social motivation. Learning more about the biological mechanisms that underlie vole social behaviors may help scientists identify fundamental aspects of social behavior that may apply to other species including humans.

specific types of social interaction, particularly access to pups, social play, and sexual opportunities (reviewed in *Trezza et al., 2011*). Social motivation has also been assessed with access to novel same-sex peers (*Martin and Iceberg, 2015*; *Achterberg et al., 2016*; *Borland et al., 2017*). Often social interactions are affiliative, but in some contexts animals will work for access to aggressive inter-actions (*Azrin et al., 1965*; *Falkner et al., 2016*; *Golden et al., 2017*). To date, only one study has examined the role of familiarity in social motivation, in novelty-preferring female rats (*Hackenberg et al., 2021*), and none have done so with mate relationships.

Prairie voles, *Microtus ochrogaster*, and meadow voles, *Microtus pennsylvanicus*, both form selective social relationships but exhibit different social organization and mating systems. Prairie voles are socially monogamous, forming long-term selective relationships between males and females that have been studied for decades (*Carter et al., 1995*; *Walum and Young, 2018*). Prairie voles also form selective relationships with familiar same-sex cage-mate 'peers' (*DeVries et al., 1997*; *Beery et al., 2018*; *Lee et al., 2019*). Meadow voles are promiscuous breeders that transition to living in social groups and sharing nests during winter (*Getz, 1972*; *Madison and Mcshea, 1987*). Under conditions of short daylength in the laboratory, female (but not male) meadow voles exhibit greater social huddling and less aggression than their long daylength counterparts (*Beery et al., 2008b*; *Lee et al., 2019*). These vole species thus allow comparison of the properties of peer relationships across species (prairie vole peers versus meadow vole peers) and relationship type within species (prairie vole mates versus prairie vole peers).

Prairie voles exhibit socially conditioned place preferences (sCPP) for familiar opposite-sex mates (*Ulloa et al., 2018*; *Goodwin et al., 2019*), and in some circumstances for same-sex peers (*Lee and Beery, 2021*). In contrast, meadow voles do not form sCPP and may even condition away from social cues (*Goodwin et al., 2019*). Neurochemical pathways underlying social reward also vary between species and relationship type; dopamine is necessary for the formation of opposite-sex pair bonds in prairie voles (*Aragona and Wang, 2009*), but is not necessary for the formation of same-sex peer preferences in meadow or prairie voles (*Beery and Zucker, 2010*; *Lee and Beery, 2021*). These initial

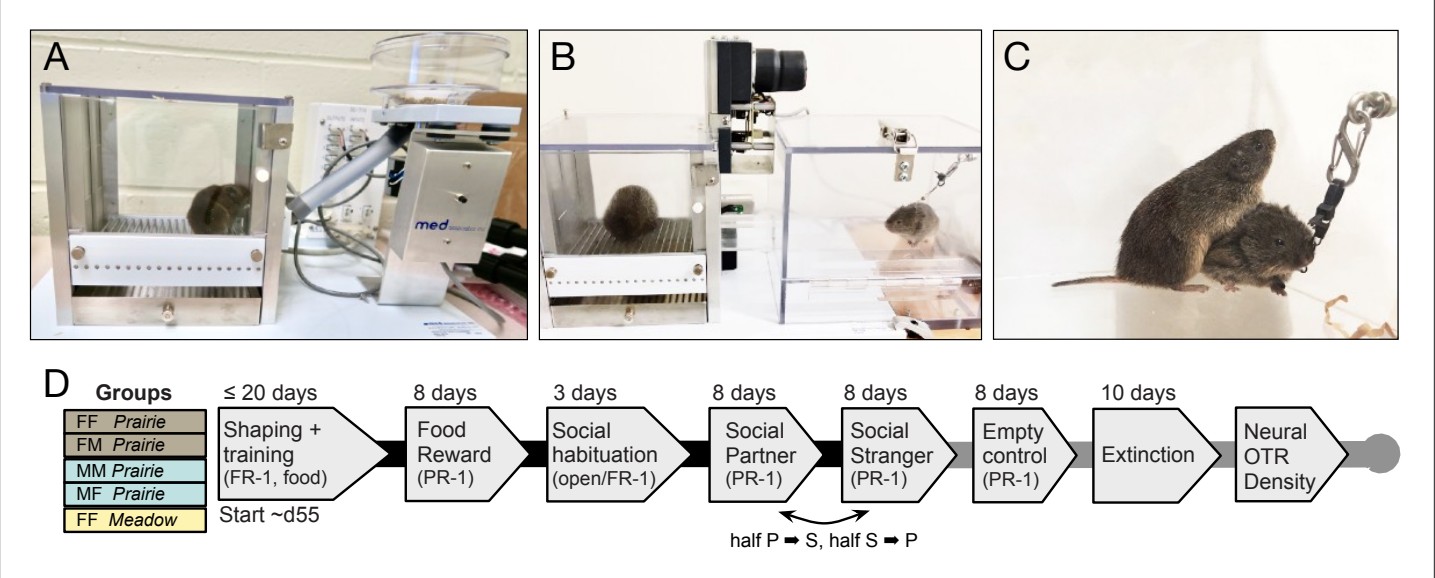

**Figure 1.** Overview of apparatuses, timeline, and testing groups. (**A**) Lever pressing in voles was shaped and trained using food reinforcement. (**B,C**) In social operant testing a lever operated a motorized door, providing 1 min access to a conspecific tethered in a connected compartment. (**D**) Five groups were tested, abbreviated here as focal sex-partner sex-species abbreviation (e.g. FF *Prairie* indicates a female prairie vole trained as a lever presser and housed with a female partner). Prairie = prairie vole (*Microtus ochrogaster*); Meadow = meadow vole (*Microtus pennsylvanicus*). Black lines connect testing phases completed by all study subjects; gray lines connect additional phases completed by a subset of subjects.

findings suggest that social selectivity may result from differential social motivation and tolerance in these species.

Voles demonstrate striking preferences for familiar versus novel peers and mates, assessed using the partner preference test (*Williams et al., 1992b*; *Beery, 2021*). This test quantifies preference, but as no effort is required to access a conspecific, it cannot distinguish between prosocial motivation and avoidance of unfamiliar conspecifics. To examine the role of motivation in relationships, we assessed effort expended by voles of different sexes (male, female), relationship types (same-sex, opposite-sex), and species (prairie vole, meadow vole) to reach social targets in an operant conditioning paradigm. Because the seasonal transition from solitary to social is most pronounced in female meadow voles in the field and laboratory (*Madison and Mcshea, 1987*; *Beery et al., 2009*), only females of this species were used. Subjects underwent >60 active training and testing days (*Figure 1*). Responses (lever presses) in lightly food-restricted voles were shaped and reinforced using a food reward, followed by 8 days of pressing for a food reward on a progressive ratio 1 (PR-1) schedule. Social testing consisted of 8 consecutive test days in which each reward consisted of 1 min of access to the familiar (same- or opposite-sex) partner, and 8 test days for which rewards consisted of access to different sex-matched strangers (order balanced within groups). We assessed effort expended to access familiar and novel social stimuli in four groups of prairie voles (*Figure 1*): females lever pressing for a female conspecific (F►F), females pressing for a male conspecific (F►M), males pressing for a male conspecific (M►M), and males pressing for a female conspecific (M►F). Meadow vole females (F►F) were also trained and tested for 8 days of familiar and 8 days of novel vole exposure, counterbalanced. A subset of voles was used to explore the reward value of an empty chamber, extinguishing timelines, and relationships between oxytocin receptor (OTR) density and behavior.

Oxytocin is involved in social recognition as well as in preference for familiar individuals (reviewed in *Anacker and Beery, 2013*), and in many instances, oxytocin signaling alters the rewarding properties of social stimuli (*Dölen et al., 2013*; *Borland et al., 2018*). We conducted receptor autoradiography to assess variation in neural OTR density in female prairie voles. (OTR was not analyzed in male brains; following early results, later males were used to pilot a two-choice social operant paradigm.)

Together these studies allowed us to examine how the reward value of social contact differs between male and female prairie voles, between opposite-sex and same-sex pairings, and between meadow and prairie vole FF pairings. We found both similarities in and striking differences between

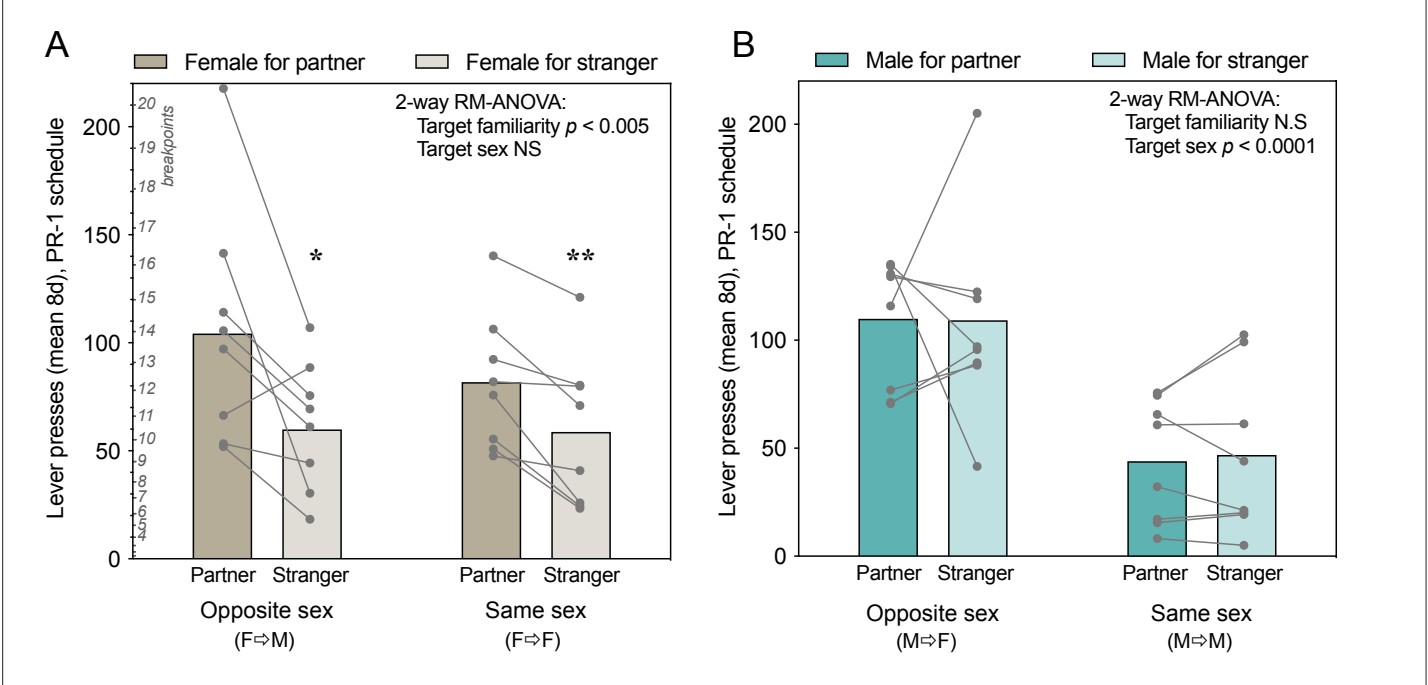

**Figure 2.** Sex-specific patterns of effort expended to access different social stimuli on a progressive ratio 1 (PR-1) schedule. (**A**) Female prairie voles responded more for familiar than unfamiliar voles of either sex. (**B**) Male prairie voles pressing for females responded more than did males pressing for males, regardless of familiarity. Dots represent mean number of responses across eight 30 min PR-1 sessions for each vole. Bars represent group means. PR-1 breakpoint thresholds are listed in italics next to the corresponding number of responses on the interior y-axis of panel A and apply to all lever pressing data (e.g. a vole that presses 55 times should receive 10 rewards, the last of which takes 10 responses to gain). Asterisks indicate significant familiarity preferences within groups (paired t-tests). *p < 0.05, **p < 0.01.

The online version of this article includes the following figure supplement(s) for figure 2:

**Figure supplement 1.** Individual lever pressing (LP) data for each prairie vole tested with a partner and stranger (8 days each).

social motivation across species, sexes, and pairing types. Detailed examination of social behaviors during social access further underscored the distinction between social motivation and familiarity preference, especially in males. In addition to these group differences in social motivation, individual differences in OTR density were related to aggressive and prosocial behaviors.

## Results

### Sex-specific patterns of social effort in prairie voles

In order to assess motivation for different kinds of social stimuli across groups, lever pressing responses were quantified on a progressive ratio schedule (PR-1). Males and females showed qualitatively different response patterns in the social chambers, as well as significant interaction between sex and variables of interest in a model screening for sex differences (sex*stimulus type (p = 0.01), sex*stimulus familiarity (p = 0.09)), so responses were further analyzed separately by sex (***Beery, 2018***; ***Beltz et al., 2019***). For each sex, two-way repeated measures ANOVA (RM-ANOVA) was performed with familiarity of the tethered stimulus (partner/stranger) as the within-subjects/repeated measure, and sex of the tethered stimulus (opposite-sex/same-sex) as a between-subjects measure. Female prairie voles pressed more for familiar partners than unfamiliar strangers, with no effect of opposite-sex versus same-sex pairings (***Figure 2A***, effect of stimulus familiarity: $F_{(1,\ 14)}$ = 15.17, p = 0.0016, $\eta_p^2$20.52; no effect of stimulus sex: $F_{(1,\ 14)}$ = 0.44, p = 0.51, $\eta_p^2$20.03; subject matching: $F_{(14,\ 14)}$ = 4.2, p = 0.0057, $\eta_p^2$20.81, no significant interaction). Paired t-tests were used for within-group comparisons of responses for the partner or stranger: familiarity preferences were significant in females paired with males ($t_{(7)}$ = –2.7, p = 0.03, d = 0.96) as well as in females paired with females ($t_{(7)}$ = –4.1, p = 0.0048, d = 1.43). The mapping from response count to the corresponding PR-1 breakpoint (i.e. the

maximum number of responses exhibited to achieve a reward) is shown in *Figure 2A* and applies to all response count figures.

Male prairie voles pressed at a higher rate for opposite-sex social stimuli regardless of familiarity (effect of stimulus sex: $F_{(1, 14)}$ = 17.4, p = 0.0009, $\eta_p^2$p20.71; no effect of familiarity, $F_{(1, 14)}$ = 0.013, p = 0.91, $\eta_p^2$p20.00, no significant effects of subject matching or interaction).

Because each vole was tested in eight consecutive sessions of each type, familiarity preference could also be assessed within individuals across days. Significant within-vole familiarity preferences were present in more female pressers (6/8 F►M and 3/8 F►F) than males (1/8 M►F and 0/8 M►M pairs) (*Figure 2—figure supplement 1*; p = 0.0059 Fisher's exact test). One male in a M►F pair exhibited a significant preference for stranger females (*Figure 2—figure supplement 1*), and mounted/copulated with strangers in multiple test sessions.

## Social motivation and behavior were parallel in female but not male prairie voles

In female prairie voles, the familiarity preference for both mates and peers in lever pressing was mirrored in cohabitation time and huddling. Even when these behaviors were scaled relative to lever presses (and thus access time), females spent a significantly higher fraction of the available time in the social chamber (time in social chamber/access time) when it was occupied by a familiar vole rather than a novel one (effect of familiarity $F_{(1,14)}$=95.06, p < 0.0001, $\eta_p^2$p20.87; subject matching $F_{(14,14)}$ = 2.789, p = 0.03, $\eta_p^2$p20.73; others NS; two-way RM-ANOVA, *Figure 3A*). Females also spent more of the

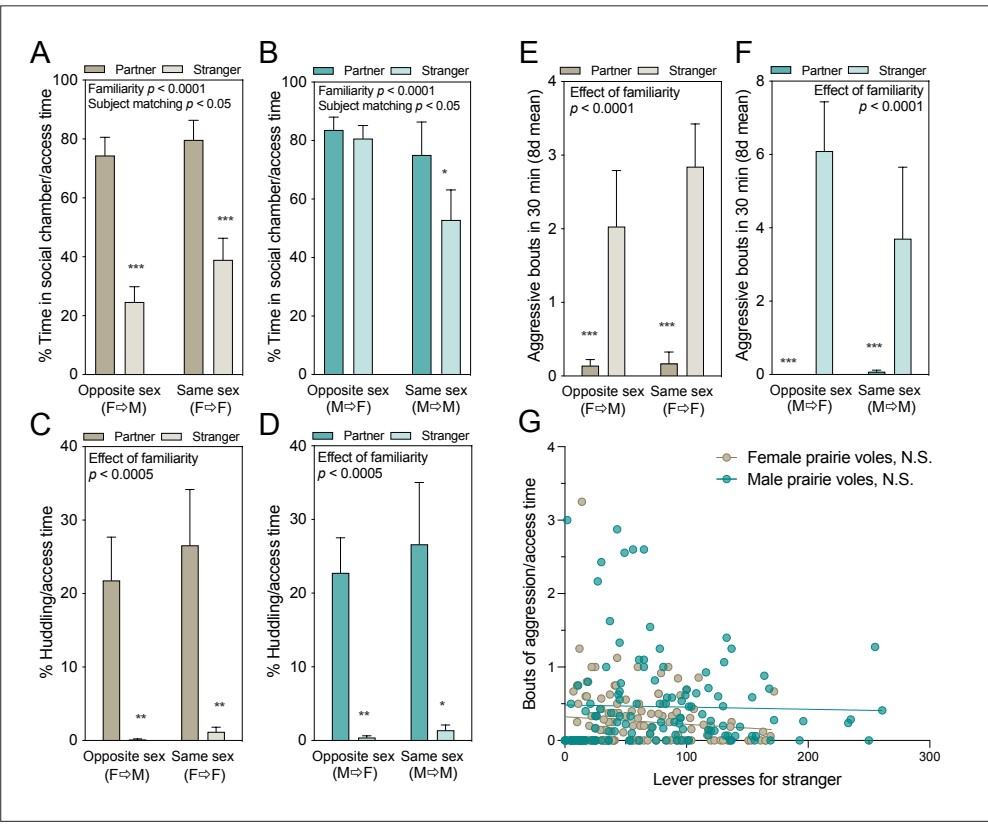

**Figure 3.** Affiliative and aggressive interactions with stimulus voles. Data represent the 8 -day testing mean for each vole (n = 8/group, ± SEM). (**A,B**) Percent of time focal voles spent in the social chamber relative to time when the door was open, allowing chamber access. Females shown in A, males in B. (**C,D**) Percent time spent huddling out of access time (i.e. when the door was raised). Significant effects of two-way repeated measures ANOVA (RM-ANOVA) are reported above each graph. Asterisks represent the results of within-groups paired t-tests. (**E,F**) Prairie voles exhibited significantly more bouts of aggression toward strangers (p < 0.0001), and there were no significant effects of sex of the presser or of the social target. (**G**) No relationship was present between daily lever pressing for access to strangers and aggression scaled by access time in male or female prairie voles. NS = not significant, *p < 0.05, **p < 0.01, ***p < 0.001.

available time huddling (time spent in immobile side-by-side contact/access time) with familiar rather than unfamiliar conspecifics of either sex (effect of familiarity: $F_{(1,14)}$ = 25.82, p = 0.0002, $\eta_p^2$p20.65; others NS; two-way RM-ANOVA, *Figure 3C*). Within-group matched comparisons of time spent with a partner or stranger also revealed that females exhibited significant familiarity preferences in time spent in the social chamber or huddling with the stimulus animal relative to time with access (time in social chamber/access time: FF: p < 0.0001, d = 3.51; FM: p = 0.0006, d = 2.12; time huddling/access time: FF: p = 0.0090, d = 1.26; FM: p = 0.0083, d = 1.29; paired t-tests).

In contrast, while males exhibited no familiarity preferences in lever pressing responses, they still exhibited strong familiarity preferences in social interaction. Males spent more of the available time in the social chamber when the tethered stimulus was familiar (effect of familiarity $F_{(1,14)}$ = 6.33, p = 0.02, $\eta_p^2$p20.31; subject matching $F_{(14,14)}$ = 4.459, p = 0.0042, $\eta_p^2$p20.24; others NS; two-way RM-ANOVA, *Figure 3B*), and huddling behavior was even more specific, with a strong effect of stimulus familiarity (partner versus stranger) and no effect of stimulus sex (opposite- versus same-sex) on the percent of [time huddling]/[time with access to the social chamber] (effect of familiarity: $F_{(1,14)}$ = 25.27, p = 0.0002, $\eta_p^2$p20.64; all else NS; *Figure 3D*). Within-group matched comparisons also revealed significant familiarity preferences in huddling time relative to access (huddling/access time: MM: p = 0.0177, d = 1.09, MF p = 0.0022, 1.66), with lesser or no familiarity preference in chamber time (time in social chamber/access time: MM: p = 0.0390, d = 0.90, MF: p = 0.56, d = 0.21; paired t-tests). There was no apparent sex difference in huddling behavior between male and female prairie voles, confirmed by pooling males and females in a three-way ANOVA (effect of focal sex NS, p = 0.91; significant effect of stimulus familiarity ($F_{(1,56)}$ = 48.03, p < 0.0001, $\eta_p^2$p20.46); effect of stimulus sex; NS, no significant interactions).

## Other social/sexual behaviors in prairie voles

Aggressive behavior was exhibited by prairie voles in all groups during social operant sessions and was analyzed by RM-ANOVA on all voles tested with partners and strangers (between-subjects factors: sex of presser (M/F)*pairing type [same/opposite sex]; within-subjects factor: target familiarity). Both males and females engaged in far more bouts of aggression with strangers than familiar partners ($F_{(1,29)}$ = 30.22, p < 0.0001, $\eta_p^2$p20.51, *Figure 3E and F*). There was no significant effect of sex of the presser ($F_{(1, 29)}$ = 3.36, p = 0.077, $\eta_p^2$p20.10), pairing type (same-sex or opposite-sex), or interactions between these variables.

Because aggression was primarily targeted at strangers, we asked whether stranger aggression might be motivating: that is, whether aggression was associated with greater lever pressing for strangers. Correlation of daily stranger lever pressing with bouts of aggression was not significant across females (R = 0.14, p = 0.10), but was significant across males (R = 0.25, p = 0.004). Because more time with access to a stranger provides more opportunity for aggression to occur, aggressive bouts were also scaled relative to access time, as was done for the other social measures. Across groups there were no relationships between stranger-directed daily lever pressing and aggression/access time in either male or females (males: R = 0.04, p = 0.64; females: R = 0.12, p = 0.16, *Figure 3G*).

Mounting behavior was present in five prairie voles, all of which were male prairie voles tested with novel (unfamiliar) female voles. This distribution was significantly non-random across the eight testing combinations used in prairie voles (e.g. male with female partner, male with female stranger, etc.) ($\chi_{(7)}^2$ = 37.97, p < 0.0001). These five voles exhibited an average of 6 bouts of mounting per testing session.

## Neural OTR density related to behavior and housing

OTR density was associated with both motivated and aggressive social behaviors in different brain regions in female prairie voles (males not assayed). There was a strong positive correlation between OTR density and lever presses for same-sex partners in the nucleus accumbens (NAcc) core (R = 0.959, p = 0.0098) and shell (R = 0.948, p = 0.0141, *Figure 4A*). There was also a strong positive correlation between mean bouts of stranger-directed aggression and OTR density in the bed nucleus of the stria terminalis (BNST) in female prairie voles (R = 0.719, p = 0.0126), again connecting receptor binding to behavior. Binding density in the BNST was not associated with stranger approach or avoidance, operationalized as time spent in the stranger's social chamber relative to access time (R = 0.350, p = 0.29), or lever presses for the stranger's chamber (R = 0.264, p = 0.43).

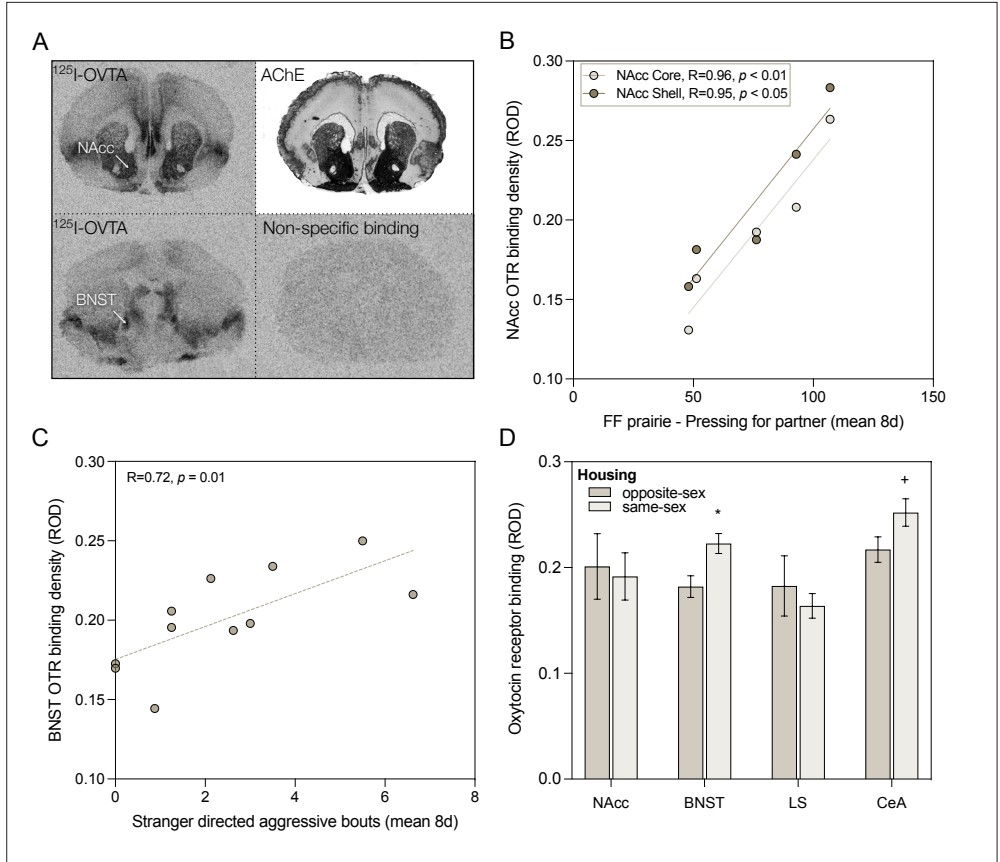

**Figure 4.** Oxytocin receptor (OTR) density, housing, and social behavior. (**A**) Representative images of $^{125}$I-OVTA binding patterns in the brains of female prairie voles. The top row shows binding at the level of the nucleus accumbens (NAcc), and an adjacent acetylcholinesterase (AChE) stained section for anatomical verifications. The bottom row shows binding at the level of the bed nucleus of the stria terminalis (BNST) and late lateral septum, as well as an adjacent section from the same brain processed for non-specific binding using the highly selective OTR antagonist [Thr$^4$Gly$^7$]-oxytocin. (**B**) OTR binding in the NAcc core and shell were strongly correlated with individual variation in lever pressing for a partner. (**C**) OTR binding in the BNST was positively correlated with stranger-directed aggression in females across pair types. (**D**) OTR density also varied in response to housing/pairing condition (opposite-sex versus same-sex). *p < 0.05, $^+$p < 0.1.

OTR density varied with housing condition. Females housed with same-sex cage-mates showed no difference in OTR density in the NAcc or lateral septum (LS), higher OTR density in the BNST (t(8.99) = 2.93, p = 0.0167, d = 1.78), and a non-significant trend in the central amygdala (t(8.71) = 1.92, p = 0.0883, d = 1.17) compared to females housed with opposite-sex cage-mates (*Figure 4C*).

## Interspecific comparisons: responses were reward-specific and comparable across species and sexes

Lever pressing responses in prairie voles were compared to those of a related non-monogamous vole species (the meadow vole) that exhibits group living during winter months. Female meadow voles are territorial and aggressive in summer or long daylengths in the lab, but socially tolerant in winter or short days. Because male meadow voles do not undergo this transition (*Madison and Mcshea, 1987*; *Beery et al., 2009*), we focused on comparison of social motivation in female meadow voles relative to female prairie voles. Prior to making this comparison, we assessed whether species and sexes differed in their lever pressing effort in response to a common reward (food). There were no sex or species differences in the number of lever pressing responses for a food reward (PR-1 schedule; 8 days averaged per subject) between female prairie voles, male prairie voles, and female meadow voles (F$_{(2,40)}$ = 1.18, p = 0.32, $\eta^2$ = 0.56; one-way ANOVA; *Figure 5A*). Food responses and social responses were converted to response rates for comparison across trials with different active lever

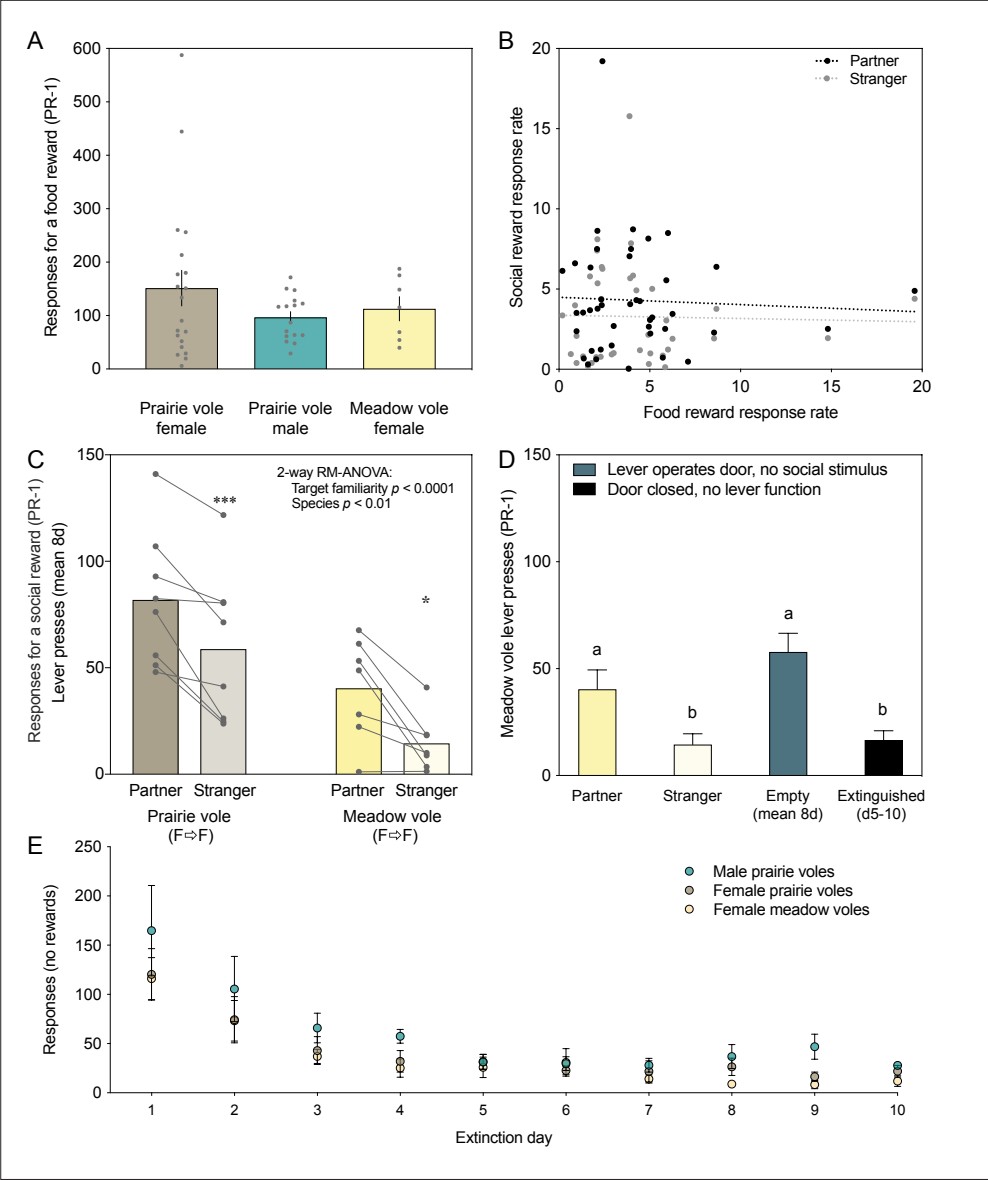

**Figure 5.** Quantifying responses across species, sexes, and reward types. (**A**) Responses for a food reward did not significantly differ between prairie voles of different sexes or between meadow and prairie vole females. Each data point represents the 8 -day mean of responses from a vole tested using a progressive ratio 1 (PR-1) schedule in 30 -min sessions. (**B**) Food response rate did not predict social response rate for familiar or unfamiliar stimuli. Data points show prairie vole response rates for food pellets on a PR-1 schedule (8 -day mean for each vole) versus social reward (black: partner; gray: stranger) on a PR-1 schedule (8 -day mean for each vole). (**C**) Meadow voles, like prairie voles, pressed more for a partner than a stranger, but pressed significantly less overall. (**D**) Social pressing for a partner in meadow voles was no higher than pressing for an empty chamber, and stranger pressing was similar to the minimum achieved by extinction. (**E**) Extinction profile over 10 days for each species and sex tested. Lever presses diminished rapidly over the first 4–5 days of testing with an inactive lever.

The online version of this article includes the following figure supplement(s) for figure 5:

**Figure supplement 1.** Individual data for each meadow vole tested with a partner and stranger (8 days each).

pressing periods: individual response rates for a food reward did not predict response rates during social testing for either the partner (p = 0.78) or the stranger (p = 0.98), indicating that responses were not subject-specific across reward types (*Figure 5B*). These findings validate the specificity of comparisons across species, sexes, and reward types.

## Meadow voles exhibited familiarity preferences but low social response rates

Female meadow voles pressed significantly more for familiar females than novel females ($t_{(6)}$=3.637, p = 0.0109, d = 1.37, paired t-test, *Figure 5C*; males not tested). This preference was individually significant within four of the seven meadow voles (*Figure 5—figure supplement 1*). Comparisons of time spent with a partner or stranger when the door was up also revealed significant familiarity preferences (P versus S for social chamber/access time: p = 0.0351, d = 1.02; P versus S for huddling/access time: p = 0.0357, d = 1.02; paired t-tests).

Despite familiarity preference, meadow vole response rate for both partners and strangers was low. Direct comparison with female prairie voles tested under the same conditions reveals that while both groups pressed more for familiar partners than for strangers, there was significantly less lever pressing in female meadow voles (two-way ANOVA, effect of target familiarity: $F_{(1,13)}$ = 29.51, p < 0.001, $\eta_p^2$ = 0.69, effect of species: $F_{(1,13)}$ = 9.71, p < 0.01, $\eta_p^2$p20.43, *Figure 5C*). Comparison of lever presses between social conditions and non-social 'empty control' conditions indicates that, for female meadow voles, the partner was not more rewarding than the empty chamber control, stranger pressing was significantly lower than empty control, and it was similar to the post-extinction level of pressing (*Figure 5D*).

## Other social/sexual behaviors in meadow voles

Aggression was rare in meadow vole trials (mean 0.3 bouts/trial), and as in our prior studies (*Lee et al., 2019*) it was significantly less frequent than aggression between female prairie voles (mean 2.3 bouts/trial, species difference: $t_{(3.83)}$, p = 0.001). No mounting behavior was observed in meadow vole tests, all of which were conducted in female voles.

## Empty chamber control and extinction

At the conclusion of social testing, all voles from cohorts 4 to 7 were tested for effort expended to explore an empty chamber without a tethered partner or stranger for 8 days each (n = 6 meadow females, 10 prairie females, and 14 prairie males). Voles were distributed across all housing types. There was no species difference in pressing for the empty chamber (meadow vole female versus prairie vole female). In both male and female prairie voles, the extent of lever pressing for the control chamber was correlated with pressing for the stranger (females: R = 0.75, p = 0.013; males: R = 0.71, p < 0.005) but not with lever pressing for the partner.

The same cohorts were then tested for extinction of lever pressing over 10 days of trials in which the door was closed and the lever did not activate the motor. All groups extinguished lever pressing behavior within ~5 days of testing (*Figure 5*). Repeated measures analysis revealed a significant effect of day of testing on pressing ($F_{(9,21)}$ = 3.72, p = 0.0063, $\eta_p^2$p20.61) but no significant effect of the testing group on extinction ($F_{(2,29)}$ = 0.76, p = 0.48, $\eta_p^2$p20.05).

## Discussion

Male and female prairie voles worked for brief access to conspecifics, but exhibited quantitatively and qualitatively different patterns of pressing, indicating striking sex differences in social motivation. In females, lever pressing effort was based on familiarity of the social target (partner versus stranger), but did not differ between same-sex (FF) and opposite-sex (FM) housed pairs. Because testing occurred with only the partner or stranger present at any given time, failure to spend time in the stranger chamber indicates lack of interest in the stranger, as opposed to relative preference for a better option. Females also exhibited extensive partner huddling and time spent in the chamber of a partner but not a stranger. These preferences persisted when scaled by rewards (i.e. time the subject was available), indicating strong selectivity in social preferences. Social motivation thus paralleled social selectivity in females.

Male prairie voles exhibited similarly strong selectivity in huddling and chamber preferences, consistent with decades of work showing partner preferences in both male and female prairie voles. In contrast, males showed no propensity to work harder to access a familiar vole than an unfamiliar social target, but instead worked significantly harder to access an opposite-sex social target than a same-sex social target. This reveals a dissociation between social motivation and markers of social

bond formation such as huddling in males. This sex difference in motivated behavior is consistent with the hypothesis that outwardly similar partner preferences in males and females result from latent differences in underlying signaling pathways (*De Vries, 2004*). Oxytocin, dopamine, and opioid signaling all affect partner preferences in males and females (*Williams et al., 1992a*; *Gingrich et al., 2000*; *Aragona et al., 2003*; *Burkett et al., 2011*; *Resendez et al., 2012*; *Johnson et al., 2016*), but prairie voles also exhibit sex differences in these signaling pathways (e.g. *Winslow et al., 1993*; *Martin et al., 2015*; *Ulloa et al., 2018*). These latent differences in mechanisms underlying social bonding may support similar partner preference behavior while promoting sex differences in social motivation.

Factors that may particularly motivate males to access unfamiliar females include opportunities for mating and aggression. Male prairie voles exhibit multiple mating strategies in field settings, including both a socially monogamous 'resident' partner strategy, and a 'wanderer' strategy; however, even residents engage in extra-pair copulations (*Madrid et al., 2020*). Interest in non-partners can also result from motivation for aggressive behavior; for example, the opportunity for aggression is rewarding in dominant male mice (reviewed in *Golden et al., 2019*). That does not seem to be the case in male prairie voles, however. Aggression toward partners was rare, and response rate was not correlated with aggression toward a stranger in male prairie voles tested with females or males. Aggressive bouts in male prairie voles tested with strangers initially appeared correlated with lever pressing effort, but this effect disappeared when scaled by access time, unlike effects reported for huddling/access time. When social interest is high (e.g. males for unfamiliar females), it is still possible that males would press more for their partners if placed in direct opposition to a stranger, and this is an avenue for future investigation.

The lack of consistent mapping between effort in the operant task and partner preferences in male huddling highlights a disconnect between social reward and the selectivity of huddling preferences. This disconnect is further underscored by the presence of robust partner preferences in female meadow voles despite no evidence of social reward in the operant task or in sCPP tests (*Goodwin et al., 2019*). Thus, partner preference does not imply social reward, nor does social reward imply selective preference. These behavioral findings are consistent with the lack of effects of dopamine antagonists on same-sex peer partner preferences in female meadow voles as well as prairie voles (*Beery and Zucker, 2010*; *Lee and Beery, 2021*). While dopamine signaling is not necessary for peer partner preference expression, it can enhance preferences (*Lee and Beery, 2021*) and may play a more fundamental role in pair bonding with mates (*Aragona and Wang, 2009*). Because partner preference does not indicate behavioral reward, the partner preference test and other tests of social approach in the absence of work likely reflect different combinations of partner tolerance, partner reward, and stranger aversion.

## OTR signaling differs by relationship type and by individual social behaviors

Strong relationships were present between OTR density, housing differences, and behavior, highlighting connections across levels of organization. Variation in OTR density by relationship type has not been previously assessed, although OTR density or mRNA levels differ in response to early-life housing manipulations in prairie voles, such as presence of a father and single versus group housing (*Prounis et al., 2015*) as well as chronic social isolation in adulthood (*Pournajafi-Nazarloo et al., 2013*).

Oxytocin signaling plays a role in diverse social behaviors in prairie voles, including pair bond formation, consolation behavior, and alloparental care (*Williams et al., 1992a*; *Olazábal and Young, 2006*; *Bales et al., 2007*; *Burkett et al., 2016*). Furthermore, oxytocin signaling has been related to social reward in non-selective mice and hamsters (*Dölen et al., 2013*; *Song et al., 2016*; *Borland et al., 2018*). Strong correlations between NAcc OTR and lever pressing for the partner in the present study provide additional support for the role of NAcc OTR in social reward. Neural OTR was related to aggressive behavior as well as prosocial behavior, underscoring the complexity of oxytocin signaling in different brain regions (*van Anders et al., 2013*; *Beery, 2015*).

## Species differences

Social pressing differed quantitatively but not qualitatively by species in meadow and prairie voles. Females of both species pressed more for partners than for strangers, but responses were lower in

meadow voles, indicative of the lack of social reward. This is consistent with prior findings from sCPP tests, in which meadow voles did not condition toward a bedding associated with social contact, and in one setting conditioned away from it (*Goodwin et al., 2019*). These findings are also in line with results from the sole prior study of operant responses in voles. *Matthews et al., 2013*, tested prairie voles and meadow voles housed in long daylengths to determine whether they would learn to lever press for stranger voles. Only prairie voles demonstrated clear learning in this scenario, consistent with low stranger interest in meadow voles housed in the long daylengths that promote territorial behavior in this species (*Beery et al., 2008b*). Nonetheless, even under pro-social short daylength conditions used in the present study, social pressing was low in meadow voles. Comparison of short daylength-housed female meadow vole responses for the partner chamber, stranger chamber, and an empty chamber in different trial blocks revealed equivalent levels of pressing for a partner or an empty chamber and less for the stranger. This suggests that decreased pressing for the stranger represents avoidance, but that pressing for the partner may indicate tolerance more than reward. Female (short daylength-housed) meadow voles also exhibited lower aggression than female prairie voles, consistent with social tolerance, and with prior descriptions of their behavior (*Lee et al., 2019*).

## Comparability across vole species and sexes

Lever pressing was demonstrated to be an effective metric to compare effort exerted to reach different social stimuli in voles; voles of each species and sex tested pressed at comparable rates for food reward, indicating a lack of major differences in task learning, and thus that social lever pressing can be assessed and compared across groups. Subject response rates were not consistently high or low across reward conditions, indicating that responses are reward-specific. Extinction was effective, with all subjects decreasing lever pressing behavior by more than half their baseline response count. Differences in lever pressing effort between groups could therefore be attributed to reward-specific differences in social motivation.

## Implications for the evolution of social relationships

Persistent relationships within specific pairs or groups of conspecifics are present throughout the animal kingdom, including species of invertebrates, fishes, amphibians, reptiles, birds, and mammals (*Bales et al., 2021*). While the nature and extent of these relationships vary considerably, they share in common the specificity of social preferences that leads to repeated association. They may differ, however, in the mechanisms that influence familiar approach and unfamiliar avoidance. In particular, familiar individuals—whether mates or peers—may or may not be socially rewarding, and unfamiliar individuals may or may not be aversive.

Even within closely related vole species, we see evidence that only some relationships involve selective social reward, for example, mate relationships in female prairie voles, while others—such as peer relationships in winter phenotype meadow voles—involve selectivity without appreciable reward. Selectivity in the absence of reward may rely instead on changing social anxiety and aggression (*Beery, 2019*). For example, when exposed to the short, winter photoperiods associated with the transition from solitary to group living in the wild, meadow voles undergo changes in CRF (corticotropin-releasing factor) receptor densities, glucocorticoid secretion, behavioral indicators of anxiety, and aggression (*Ossenkopp et al., 2005*; *Beery et al., 2014*; *Anacker et al., 2016*). More research is needed to establish causal links between these changes and the transition to group living. More broadly, it remains to be determined to what extent social monogamy and pair bonding with mates shares mechanisms across species (*Goodson, 2013*), and to what extent different types of relationships (e.g. with peers or mates) share foundations, or differ in their regulation. Ultimately, these studies should help us understand how selective relationships of different types evolve.

## Conclusions

While other studies have assessed social reward in rodents, few have considered the role of stimulus familiarity, likely because laboratory rodents do not exhibit familiarity preferences under normal conditions (reviewed in *Beery and Shambaugh, 2021*). In social choice tests, mice and young rats often prefer social novelty (*Moy et al., 2004*; *Smith et al., 2015*), and relative preference for a social stimulus versus a food stimulus is greater when novel rats are presented (*Reppucci et al., 2020*). Indeed, in operant trials in which rats had simultaneous access to familiar and unfamiliar same-sex

conspecifics, rats expended more effort to access unfamiliar conspecifics (*Hackenberg et al., 2021*). In the present study, female prairie voles exhibited similar partner preferences but higher social motivation and aggression compared to female meadow voles. Social motivation and selectivity were not linked in male prairie voles, and there was a striking sex difference in the reward value of mates and peers in prairie voles. OTR binding revealed connections between social environment, receptor density, and prosocial behavior, illustrating the importance of this system across levels of biological organization. Better understanding of the interface between social motivation and social selectivity will thus be key to improving our understanding of the nature of social relationships.

## Materials and methods

### Key resources table

| Reagent type (species) or resource | Designation | Source or reference | Identifiers | Additional information |
|---|---|---|---|---|
| Chemical compound, drug | (Thr$^4$,Gly$^7$)-Oxytocin | Bachem | 4013837 | |
| Chemical compound, drug | $^{125}$I-OVTA; $^{125}$I-ornithine vasotocin analog; vasotocin, d(CH$_2$)$_5$ [Tyr(Me)$_2$,Thr$^4$,Orn$^8$,($^{125}$I)Tyr$^9$-NH$_2$] | Perkin Elmer | NEX254050UC | |
| Chemical compound, drug | Testosterone | Sigma-Aldrich | T1500 | |
| Software, algorithm | MED-PC IV | Med Associates | SOF-735 | |

### Animal subjects

Prairie voles and meadow voles from in-house colonies were bred in a long photoperiod (14 hr light:10 hr dark; lights off at 17:00 EST; described further in *Lee et al., 2019*). Meadow voles were weaned into the winter-like short photoperiods associated with group living in this species (10:14 light:dark; lights off at 17:00 EST). Voles were pair-housed in clear plastic cages with aspen bedding and an opaque plastic hiding tube. Food (5015 supplemented with rabbit chow; LabDiet, St Louis, MO) and water were provided ad libitum, except during food restriction (described below). All procedures adhered to federal and institutional guidelines and were approved by the Institutional Animal Care and Use Committee at Smith College.

### Timeline and groups

Training began in adulthood at 62 ± 1.3 days of age (mean ± SEM, range 41–76). Operant conditioning training and testing consisted of multiple phases described briefly here and in greater detail in subsequent sections. Responses (lever presses) were shaped and trained using a food reward on a fixed ratio 1 (FR-1) schedule. Animals that met training criteria progressed to the experimental testing sequence, beginning with 8 days of pressing for a food reward on a PR-1 schedule (*Figure 1*). Subjects in opposite-sex pairs were placed with either a tubally ligated, hormonally intact female mate, or a castrated and testosterone implanted male mate 5–10 days prior to the start of social habituation and testing. Subjects in same-sex pairs remained with their cage-mate. Social testing consisted of 8 days of PR-1 with rewards yielding access to the familiar (same- or opposite-sex) partner, and 8 days with access to a sex-matched stranger (order balanced within groups). Voles were trained and tested over seven cohorts; group membership was distributed across cohorts, and voles were assigned to groups within sex without knowledge of their response rates in the training phase. A subset of voles (those in cohorts 4–7) continued in empty chamber control and/or extinguishing tests as described below. Voles were sacrificed at the conclusion of testing, and brains were stored at –80°C.

We tested four groups of prairie voles (*Figure 1*): females lever pressing for a female conspecific (F►F), females pressing for a male conspecific (F►M), males pressing for a male conspecific (M►M), and males pressing for a female conspecific (M►F). Each group consisted of eight focal voles, tested for 8 days with their partner and for 8 days with a series of novel 'strangers', sex-matched to the partner. The order of testing (partner then stranger or stranger then partner) was counterbalanced within groups. Some voles did not complete both partner and stranger testing, in which case additional voles were added up to 8/group. Meadow vole females (F►F-*Mp*, n = 7) were also trained and tested for 8 days of familiar and 8 days of novel vole exposure, with order counterbalanced within the group.

## Operant conditioning and testing with food reward

Subjects were weighed for 3 consecutive days to establish baseline body weights, then food-restricted to a target weight of 90 % baseline to enhance motivation for the food reward. Weights were recorded daily after training or testing, prior to being returned to their home-cages. Any vole that dropped to or below 85 % of the baseline weight was returned to ad libitum food to avoid long-term health consequences. Perforated cage dividers were used during food restriction to ensure each vole had access to its specific ration (0.3–1 food pellets and ~4 g [half] of a baby carrot). Food restriction ended when subjects transitioned to social testing.

Operant conditioning was conducted in mouse-sized modular test chambers (30.5 cm × 24.1 cm × 21.0 cm) outfitted with a response lever, clicker, modular pellet dispenser for mouse, and pellet receptacle (Med Associates Inc, St Albans, VT, *Figure 1A*). Data were acquired using the MED-PC-IV program running training protocols coded by experimenters. Sessions lasted 30 min and took place between 0900 and 1700. Vole behavior was shaped using manual reinforcement by an experimenter until a subject met the training criterion of 3 days in a row of ≥5 responses without manual reinforcement on an FR-1 schedule. One 20 mg food pellet (Dustless Precision Pellet Rodent Grain Based Diet; Bio-Serv, Flemington, NJ) was dispensed as each reward. Animals that did not learn to consistently lever press within ~20 days were used as partners or strangers for future social testing. Subjects that met the training criterion transitioned to a PR-1 schedule with each successive reward requiring an additional response. The progressive ratio has been shown to be a better indicator of motivation than FR programs (*Hodos and Kalman, 1963*; *Weatherly et al., 2003*). PR-1 testing was conducted for 8 days, at the conclusion of which all focal animals were returned to ad lib food, and cage dividers were removed.

## Testing with social rewards

Social reward testing was conducted in mouse-sized modular test chambers, custom-equipped with a motorized door (Med Associates Inc, St Albans, VT) for access to a second 'social' chamber (*Figure 1B*). This chamber was constructed of clear plastic (15 cm × 20.5 cm × 13 cm) and contained an eye-bolt for tethering a stimulus vole (*Figure 1C*). A clear plastic tunnel (2.54 cm diameter, 5.5 cm long) connected the operant chamber to the social chamber, and the entire apparatus was fixed to a mounting board. Lever presses were rewarded by door opening and chamber access; the door remained raised for 1 min, after which the experimenter returned the focal vole to the operant chamber. Sessions lasted 30 min and were video-recorded for quantification of additional behaviors.

Subjects transitioned to social testing following a habituation session and two FR-1 sessions. Habituation to the social apparatus took place with the door open and the lever covered: voles explored the apparatus for 15 min with an empty social chamber, and 15 min with the partner tethered in the social chamber. Two days of FR-1 pressing for a tethered vole followed habituation to ensure that subjects associated lever pressing with access to the social chamber and a stimulus vole.

Social testing took part in two phases: pressing for a partner vole on a PR-1 schedule and pressing for a stranger on a PR-1 schedule. Each phase lasted 8 days. The order of testing was counterbalanced within groups and subjects completed both phases. Social stimulus animals were tethered to the end of the social chamber. During the 8 days of stranger testing, the focal vole was tested against a novel vole each session to prevent familiarity between conspecifics.

## Non-social conditions

Empty chamber testing took place after social testing to avoid altering lever pressing for the social stimuli. The empty chamber control was run to assess the value of apparatus exploration: 30 voles (10 female prairie voles, 14 male prairie voles, 6 female meadow voles) pressed the lever for 8 successive days on a PR-1 schedule to access the adjacent chamber when no stimulus vole was present. Sessions lasted 30 min and video was recorded and scored for behavior after testing. For the extinction phase, 31 voles (13 female prairie voles, 11 male prairie voles, 7 female meadow voles) were tested in the social chamber with an unrewarded lever for 10 successive days (30 min sessions).

## Behavioral scoring

Counts of responses (lever presses) and rewards (food pellets or door raises) were automatically recorded during each test. In all social trials (16/vole) and all empty chamber control trials (8/vole),

behavior in the 'social' chamber was also filmed with a portable digital video camera. Videos were scored using a custom perl script (OperantSocialTimer; https://github.com/BeeryLab/Operant/, *Beery, 2017*) to determine time in the social chamber, time in side-by-side contact with the tethered vole (huddling), and bouts of aggression. These values could also be reported relative to other intervals (e.g. time huddling/access time when the door was up, or time huddling/time in the social chamber). Non-social/empty chamber trials (8 days/vole) were also videotaped and analyzed for time in the social chamber/available time with the door raised.

## Castration and tubal ligation

At least 1 week prior to pairing, the future 'partner' of each opposite-sex prairie vole pair was surgically altered to prevent pregnancies during testing. Female partners of male focal voles underwent tubal ligation. Dorsal incisions were made over each ovary. Two knots were placed below each ovary at the top of the uterine horn. The wound was closed using a sterile suture. Male partners of female focal voles were castrated and implanted with testosterone capsules. Testes were accessed by midline incision, and the blood supply was cut-off through a tie at the testicular artery. Testes were removed and the muscle wall and skin were closed using sterile suture. A testosterone capsule was implanted subcutaneously between the scapulae. Capsules contained 4 mm of crystalline testosterone (Sigma-Aldrich, St Louis, MO) in silastic tubing (ID 1.98 mm, OD 3.18 mm; Dow Corning, Midland, MO) as in *Costantini et al., 2007*. Capsules were sealed with silicone, dried, and soaked in saline for 24 hr prior to insertion. A subset of strangers was also castrated or ligated, with no effect on focal behavior. Surgical procedures were performed under isoflurane anesthesia. Voles received 0.05 mg/kg buprenorphine and 1.0 mg/kg metacam subcutaneously prior to surgery, and again the following day. Post-operative wound checks continued for up to 10 days post-surgery.

## Receptor autoradiography

OTR binding density was assessed in the brains of 11 female prairie voles at the conclusion of the study (males were used for an additional pilot study). Frozen brains were sectioned coronally at 20 μm, thaw-mounted on Super-frost Plus slides (Fisher, Inc), and stored at –80°C until processing (as in *Beery et al., 2008a*; *Beery and Zucker, 2010*; *Mooney et al., 2015*). Briefly, slides were thawed until dry, then fixed for 2 min in fresh, chilled 0.1 % paraformaldehyde in 0.1 M PBS. Sections were rinsed 2 × 10 min in 50 mM Tris (pH 7.4), and incubated for 60 min at room temperature in a solution (50 mM Tris, 10 mM $MgCl_2$, 0.1 % BSA, 0.05 % bacitracin, 50 pM radioligand) containing the radioactively labeled $^{125}$I-ornithine vasotocin analog vasotocin, d($CH_2$)$_5$ [Tyr(Me)$_2$,Thr$^4$,Orn$^8$,($^{125}$I)Tyr$^9$-NH$_2$] ($^{125}$I-OVTA, PerkinElmer, Inc). An adjacent series of slides, processed for non-specific binding, was incubated with an additional 50 nM non-radioactive ligand [Thr$^4$Gly$^7$]-oxytocin (Bachem). All slides were rinsed 3 × 5 min in chilled Tris–$MgCl_2$ (50 mM Tris, 10 mM $MgCl_2$, pH 7.4), dipped in cold distilled water, and air-dried. Sections were apposed to Kodak BioMax MR film (Kodak, Rochester, NY) for 3 days and subsequently developed. Radioligand binding density in each brain region was quantified in samples of uniform area from three adjacent sections for each brain region and averaged for each brain. Non-specific binding was subtracted from total binding to yield specific binding values.

## Statistical analyses

Social data were analyzed for all subjects completing both partner and stranger phases of testing (n = 8 prairie vole M➤M pairs, 8 prairie vole M➤F pairs, 8 prairie vole F➤F pairs, 8 prairie voles F➤M pairs, and 7 meadow vole F➤F pairs). Four additional female prairie voles completed testing with a partner or stranger only: data from these subjects was included in analysis of food responses and food versus social response rates. Group differences in single variables (e.g. food responses) were assessed by one-way ANOVA. Two-way RM-ANOVA was used to assess the effects of social factors, with stimulus familiarity [partner, stranger] as a within-subjects repeated measure, stimulus type [same-sex, opposite-sex] as a between-subjects (non-repeated) measure, a test for interaction effects [stimulus familiarity*stimulus type], and for subject matching. Paired t-tests were used within groups for comparison of behavior toward the partner versus stranger. Response count (i.e. lever presses) and breakpoint (i.e. number of rewards achieved) are highly correlated; detailed results are therefore shown for only one measure (response count). Response rate (responses/active session time) was used when comparing food responses to social responses, as the lever

was continuously active during food-rewarded testing (active session time = 30 min), but was not capable of raising the door when it was already up (active session time = 30 min with the door up). Autoradiography data were collected in multiple brain regions, and comparisons were performed by two-way ANOVA (group*brain region). Statistical analyses were performed in JMP 15.0 (SAS, Inc) and Prism 9 (GraphPad Software Inc). Effect sizes were calculated in Excel. Cohen's d for paired t-tests used the mean of partner-stranger differences/standard deviation of partner-stranger differences. Eta squared ($\eta^2$) and partial eta squared ($\eta_p^2$) were reported for one-way and two-way ANOVAs, respectively (*Lakens, 2013*). Pearson's product-moment correlation coefficient was reported for correlations. All tests were two-tailed, and results were deemed significant at p < 0.05. Number of animals: social operant studies in rats have been successful with six subjects (*Tan and Hackenberg, 2016*; *Hiura et al., 2018*; *Hackenberg et al., 2021*). We used 30 % more subjects as a buffer ( eight females or males in each condition), as operant behavior in voles was not well characterized.

## Acknowledgements

We are extremely grateful to Dr Tim Hackenberg for advising us on the design, physical setup, and analysis of the studies described here. Lab members Emily Halstead, Amelia Windorski, Rose Hatem, Madeleine Lerner, and Marcela Rodrigues-Guimaraes assisted with behavioral testing and video scoring, and Karina Lieb assisted with brain tissue preparation. Dale Renfrow (Smith Center for Design and Fabrication) assisted with building social testing chambers. We thank the staff of the Smith College Animal Care Facility for animal care and colony maintenance. This research was supported by the National Institute of Mental Health of the National Institutes of Health under Award Number R15MH113085. Publication made possible in part by support from the Berkeley Research Impact Initiative (BRII) sponsored by the UC Berkeley Library.

## Additional information

### Funding

| Funder | Grant reference number | Author |
| --- | --- | --- |
| National Institutes of Health | R15MH113085 | Annaliese K Beery |

The funders had no role in study design, data collection and interpretation, or the decision to submit the work for publication.

### Author contributions

Annaliese K Beery, Conceptualization, Data curation, Formal analysis, Funding acquisition, Investigation, Methodology, Project administration, Supervision, Validation, Visualization, Writing – original draft, Writing – review and editing; Sarah A Lopez, Data curation, Formal analysis, Investigation, Methodology, Writing – original draft, Writing – review and editing; Katrina L Blandino, Investigation, Methodology, Writing – review and editing; Nicole S Lee, Data curation, Formal analysis, Investigation, Writing – review and editing; Natalie S Bourdon, Investigation, Writing – review and editing

### Author ORCIDs

Annaliese K Beery  http://orcid.org/0000-0002-1249-9182

### Ethics

This study was carried out in accordance with the recommendations of the Guide for the Care and Use of Laboratory Animals of the National Institutes of Health. Animals were handled according to a research protocol (ASAF 272) approved by the Institutional Care and use committee of Smith College.

### Decision letter and Author response

Decision letter https://doi.org/10.7554/eLife.72684.sa1
Author response https://doi.org/10.7554/eLife.72684.sa2

## Additional files

### Supplementary files
• Transparent reporting form

### Data availability
Data have been deposited in a project folder on the Open Science Framework website, available at: https://osf.io/g2jf7/.

The following dataset was generated:

| Author(s) | Year | Dataset title | Dataset URL | Database and Identifier |
|---|---|---|---|---|
| Beery A | 2021 | Social selectivity and social motivation in voles | https://osf.io/g2jf7/ | Open Science Framework, g2jf7 |

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
