## [Decision Letter]

**Acceptance summary:**

This paper introduces a new method to measure motivation to engage with familiar or unfamiliar individuals in prairie voles, a widely used animal model system for studying social relationships. The authors show that females will work harder to access familiar individuals (either pair-bonded males or same-sex females) while males will work to access females regardless of whether they are pair-bonded or unfamiliar. These results cast a new light on decades of work based partner-preference tests that assess pair bonds without considering motivation.

**Decision letter after peer review:**

Thank you for submitting your article "Social selectivity and social motivation in voles" for consideration by *eLife*. Your article has been reviewed by 3 peer reviewers, one of whom is a member of our Board of Reviewing Editors, and the evaluation has been overseen by Michael Taffe as the Senior Editor. The following individual involved in review of your submission has agreed to reveal their identity: Karen Bales (Reviewer #3).

Overall, the reviewers were enthusiastic about the application of operant methods to study attachments in the different species of voles.

Essential revisions:

1) The lack of data using a progressive ratio schedule was viewed as a weakness for evaluating sex differences in motivation. If these data can be collected the reviewers agreed that this would significantly strengthen the claims in the manuscript. If these data can not be collected quickly the authors should address this limitation in the discussion.

2) The absence of male oxytocin receptor autoradiography data was also viewed as a weakness. If these data are available (null or otherwise) they should be included. The genotyping data were viewed as being tangential to the main theme of the paper and reviewers agreed these data should be removed.

*Reviewer #1 (Recommendations for the authors):*

My main suggestion is for the authors to highlight how their results change how we view the results of partner preference tests in the abstract and discussion. I think it would be helpful for the authors to go into a little more depth about the difference between actively seeking out a partner (or female) versus more passively huddling (in the case of animals with an opposite sex partner) when the partner is available. Do the authors expect dopaminergic mechanisms to play a stronger role in lever pressing than huddling (similar to liking/wanting hypotheses)?

*Reviewer #2 (Recommendations for the authors):*

Many of my concerns were listed in the public review.

Including the missing male groups would improve the paper.

I am not sure the value of the genotype comparisons given the limited behavioral data that can be assessed. It might strengthen the paper to remove it (and perhaps the data could serve as the basis for a different paper).

It would be great to run progressive ratio on a subset of subjects where differences were found. For example, run progressive ratio for female prairie voles with a partner that is familiar vs. novel (e.g., female partners only) and male prairie voles for a partner that is male vs. female (e.g., familiar only). I think this could be done with relatively few animals but would allow for conclusions about motivation.

The claim "sex-specific" should be limited to when an analysis comparing sexes was actually conducted (e.g., the first title of the Results section is inaccurate).

The corresponding correlation in males for Figure 3G should be shown.

*Reviewer #3 (Recommendations for the authors):*

In general, I found the paper to be quite well-written and I enjoyed reading it. I have the following recommendations:

Because some sample sizes were relatively small, the chance of Type II error is higher. Reporting effect sizes would be helpful.

Caption for Figure 5a: should specify that this statement refers to a food reward.

Line 296: In Ahern and Young 2009, I do not believe that there were actually any differences in OTR binding due to early experience. I couldn't check the other two references for that statement because neither was actually in the reference list (see below).

Check reference list and text for correspondence: Pournajafi-Nazarloo 2013 missing from ref list; Prounis also missing; there may be more, those are just the two I noticed.

Line 377-378: "infertile but sexually active" is odd and imprecise wording.

Finally, I felt like the discussion could be broader in its consideration of the implications of these findings for the evolution of social bonding.

---

## [Author Response]

Essential revisions:1) The lack of data using a progressive ratio schedule was viewed as a weakness for evaluating sex differences in motivation. If these data can be collected the reviewers agreed that this would significantly strengthen the claims in the manuscript. If these data can not be collected quickly the authors should address this limitation in the discussion.

We agree that a progressive ratio schedule is preferable for evaluating differences in motivation, and this is the schedule already used throughout the study. We also cite references supporting the preferability of progressive ratio schedules in our description of the methods. A fixed ratio schedule was used in some parts of training (initial introduction to food rewards and habituation to the social testing paradigm), but a progressive ratio was used for all experimental phases.

We have identified two ways to make this information more accessible throughout the manuscript.

1. The testing protocol was described in greatest detail in the methods section which appears *after* the Results section. We have thus added text highlighting the progressive ratio protocol in the Results section, which now begins with: “In order to assess motivation for different kinds of social stimuli across groups, lever pressing responses were quantified on a progressive ratio schedule (PR-1).” We have also added this information to the caption of figure 2 and the y-axis of figures 2 and 5.

2. Progressive ratio results can be reported as breakpoints (which only represent PR designs) or lever pressing responses (which can be used for PR or FR studies), but as these metrics are highly correlated, only one is typically analyzed. We reported lever pressing responses as it captures additional information from incomplete reward series (detailed circa line 720), so to aid in interpreting/translating the data in terms of breakpoints, we have added the mapping of lever presses to breakpoints for a PR-1 schedule to the y-axis of Figure 2A. This is also now articulated in the text: “The mapping from response count to the corresponding PR-1 breakpoint (i.e. the maximum number of responses exhibited to achieve a reward) is shown in figure 2A and applies to all response count figures.” As well as in the figure 2 legend: “PR-1 breakpoint thresholds are listed in italics next to the corresponding number of responses on the interior y-axis of panel A and apply to all lever pressing data (e.g. a vole that presses 55 times should receive 10 rewards, the last of which takes 10 responses to gain).”

2) The absence of male oxytocin receptor autoradiography data was also viewed as a weakness. If these data are available (null or otherwise) they should be included. The genotyping data were viewed as being tangential to the main theme of the paper and reviewers agreed these data should be removed.

Male oxytocin receptor data is not available. When we discovered males were not working harder to access familiar females, we diverted later study males to pilot a social choice variant of the operant setup (and thus stopped collecting their brains at the conclusion of the study). That pilot led us to conduct a new study using a two-choice operant apparatus (now in review), and the tissues from that second study could enable a similar analysis in both male and female brains. I will strongly consider running that assay once I have autoradiography set up in my new lab.

We have clarified that male data were not omitted null data, but rather not assayed: “We conducted receptor autoradiography to assess variation in neural oxytocin receptor density in female prairie voles. (OTR was not analyzed in male brains; following early results, later males were used to pilot a two-choice social operant paradigm).”

We have removed all genotyping data as requested.

Reviewer #1 (Recommendations for the authors):My main suggestion is for the authors to highlight how their results change how we view the results of partner preference tests in the abstract and discussion. I think it would be helpful for the authors to go into a little more depth about the difference between actively seeking out a partner (or female) versus more passively huddling (in the case of animals with an opposite sex partner) when the partner is available. Do the authors expect dopaminergic mechanisms to play a stronger role in lever pressing than huddling (similar to liking/wanting hypotheses)?

Thank you for this suggestion – we have added material to both the abstract and discussion.

Abstract: we were at the word limit, so we cut a few words and added a minimal statement:

“This reveals a striking sex difference in pathways underlying social monogamy, and demonstrates a fundamental disconnect between motivation and social selectivity in males—a distinction not detected by the partner preference test.”

Discussion (new paragraph):

“The lack of consistent mapping between effort in the operant task and partner preferences in male huddling highlights a disconnect between social reward and the selectivity of huddling preferences. This disconnect is further underscored by the presence of robust partner preferences in female meadow voles despite no evidence of social reward in the operant task or in socially conditioned place preference tests (Goodwin et al., 2019). Social reward thus does not imply selective preference, nor does partner preference imply social reward. These behavioral findings are consistent with the lack of effects of dopamine antagonists on same-sex peer partner preferences in female meadow voles as well as prairie voles (Beery and Zucker, 2010; Lee and Beery, 2021). While dopamine signaling is not necessary for peer partner preference expression, it can enhance preferences (Lee and Beery, 2021), and may play a more fundamental role in pair bonding with mates (Aragona and Wang, 2009). Because partner preference does not indicate behavioral reward, the partner preference test and other tests of social approach in the absence of work likely reflect different combinations of partner tolerance, partner reward, and stranger aversion.”

We have also added a related section to the discussion “Implications for the evolution of social relationships” (see response to reviewer #3).

Reviewer #2 (Recommendations for the authors):Including the missing male groups would improve the paper.

I have added better explanations for their omission in these two instances. In the case of meadow voles, the seasonal transition to sociality only occurs in females (or predominantly in females) in both the field and laboratory, thus the following text has been added:

Introduction: “Because the seasonal transition from solitary to social is most pronounced in female meadow voles in the field and laboratory (Madison and McShea, 1987; Beery et al., 2009), only females of this species were used.”

Results: “Lever pressing responses in prairie voles were compared to those of a related non-monogamous vole species (the meadow vole) that exhibits group living during winter months. Female meadow voles are territorial and aggressive in summer or long day lengths in the lab, but socially tolerant in winter or short days. Because male meadow voles do not undergo this transition (Madison and McShea, 1987; Beery et al., 2009), we focused on comparison of social motivation in female meadow voles relative to female prairie voles.”

Regarding neural OTR data: **“**We conducted receptor autoradiography to assess variation in neural oxytocin receptor density in female prairie voles. OTR was not analyzed in male brains as several male subjects were used to pilot a follow-up study using a two-choice social operant paradigm (*in review*).”

In prior studies in voles in our lab and others, sex differences in OTR density have not been found (e.g. Insel et al., 1992, Ahern et al., 2021), with signaling differences hypothesized to be largely mediated through sex differences in peptide distribution. Nonetheless it would have been informative to correlate individual binding data with behavior in the males, and we will consider this for a second study we conducted that yielded more male brains.

I am not sure the value of the genotype comparisons given the limited behavioral data that can be assessed. It might strengthen the paper to remove it (and perhaps the data could serve as the basis for a different paper).

We have removed the genotype comparisons as suggested. They are stronger with a larger sample set, so we have added them to our second study data set for which we also collected genotype data.

It would be great to run progressive ratio on a subset of subjects where differences were found. For example, run progressive ratio for female prairie voles with a partner that is familiar vs. novel (e.g., female partners only) and male prairie voles for a partner that is male vs. female (e.g., familiar only). I think this could be done with relatively few animals but would allow for conclusions about motivation.

A progressive ratio was used for all testing. This has been clarified in several locations in the manuscript and hopefully addresses this concern.

The claim "sex-specific" should be limited to when an analysis comparing sexes was actually conducted (e.g., the first title of the Results section is inaccurate).

We have now added quantitative comparison of the sexes to supplement the qualitative description of the difference in response pattern. Specifically, we screened for sex differences using a generalized linear model with stimulus type (same/opposite sex), stimulus familiarity (partner/stranger), sex of the presser (female/male), sex of the presser*stimulus type, and sex of the presser*stimulus familiarity. This yielded interactions of sex with stimulus type (*p*=.0112) and familiarity (*p*=0.09), providing quantitative evidence for sex differences not in the amount but in the *stimulus specificity* of pressing, supporting the decision to analyze the results separately by sex. We have added this result to the manuscript. We believe “Sex-specific patterns of social effort in prairie voles” is therefore accurate. If needed, this section could be re-titled “Social effort in males and females was influenced by different aspects of the social stimulus.”

The corresponding correlation in males for Figure 3G should be shown.

Both male and female data now appear in Figure 3G. We also took this opportunity to change the graph from total bouts of aggression to aggression/access time, as was already displayed for all other social comparisons (e.g. huddling/access time, social chamber/access time). We originally presented both versions in the text, emphasized the scaled findings in the text, but displayed the unscaled findings in the figure. We continue to present both unscaled and scaled findings in the text, but now display the scaled findings we believe are more meaningful.

In addition, we added further detail regarding which voles underwent control and extinguishing testing (all voles in cohorts 4, 5, 6, and 7, which represented >2/3 of the voles in the study and were distributed across groups).

Reviewer #3 (Recommendations for the authors):In general, I found the paper to be quite well-written and I enjoyed reading it. I have the following recommendations:Because some sample sizes were relatively small, the chance of Type II error is higher. Reporting effect sizes would be helpful.

We have added effect sizes throughout the manuscript. For pairwise comparisons we have added Cohen’s d; for 1-way and 2-way ANOVAs we now present eta squared and partial eta squared, and for correlations we already listed the Pearson product-moment correlation coefficient alongside statistical significance. We selected the sample sizes for this study in consultation with operant expert Dr. Tim Hackenberg, who assured us that the repeated testing involved in social operant studies allowed him to use samples sizes as small as 4-6 per group (we used 8). Although only 8 subjects were tested per group, each subject’s data represented the mean of 8 days of sampling in each of the three conditions, enhancing the reliability of data for each subject and reducing the chance of type II error.

Caption for Figure 5a: should specify that this statement refers to a food reward.

Fixed.

Line 296: In Ahern and Young 2009, I do not believe that there were actually any differences in OTR binding due to early experience. I couldn't check the other two references for that statement because neither was actually in the reference list (see below).

Thank you for catching this – it looks like the difference in Ahern and Young 2009 was in OT, not OTR, so this reference has been removed. The other two references show similar findings, and I’ve added more study-specific detail so this now reads:

“Variation in OTR density by relationship type has not been previously assessed, although oxytocin receptor density or mRNA levels differ in response to early-life housing manipulations in prairie voles, such as presence of a father and single versus group housing (Prounis et al., 2015) as well as chronic social isolation in adulthood (Pournajafi-Nazarloo et al., 2013).”

Check reference list and text for correspondence: Pournajafi-Nazarloo 2013 missing from ref list; Prounis also missing; there may be more, those are just the two I noticed

These references have also been added to the bibliography, and one other that was also previously missing. A few other references appeared to be missing from the LaTeX formatted file that are present in the Word file for reasons I cannot explain, so I have included all revisions on the Word version of the file.

Line 377-378: "infertile but sexually active" is odd and imprecise wording.

Agreed. In lieu of finding better language that could apply to both males and females, surgical manipulations of males and females are now specified separately:

“Subjects in opposite-sex pairs were placed with either a tubally ligated, hormonally intact female mate, or a castrated and testosterone implanted male mate”

Finally, I felt like the discussion could be broader in its consideration of the implications of these findings for the evolution of social bonding.

We have added a section at the end of the Discussion section:

Implications for the evolution of social relationships

Persistent relationships within specific pairs or groups of conspecifics are present throughout the animal kingdom, including select invertebrates, fishes, reptiles and amphibians, birds, and mammals (Bales et al., 2021). While the nature and extent of these relationships vary considerably, they share in common the specificity of social preferences that leads to repeated association. They may differ, however, in the mechanisms that influence familiar approach and unfamiliar avoidance. In particular, familiar individuals—whether mates or peers—may or may not be socially rewarding, and unfamiliar individuals may or may not be aversive.

Even within closely related vole species, we see evidence that only some relationships involve selective social reward, for example mate relationships in female prairie voles, while others—such as peer relationships in winter phenotype meadow voles—involve selectivity without appreciable reward. Selectivity in the absence of reward may rely instead on changing social anxiety and aggression (Beery, 2019). For example, when exposed to the short, winter photoperiods associated with the transition from solitary to group living in the wild, meadow voles undergo changes in CRF receptor densities, glucocorticoid secretion, behavioral indicators of anxiety, and aggression (Ossenkopp et al., 2005; Beery et al., 2014; Anacker et al., 2016). More research is needed to establish causal links between these changes and the transition to group living. More broadly, it remains to be determined to what extent social monogamy and pair bonding with mates shares mechanisms across species (Goodson, 2013), and to what extent different types of relationships (e.g. with peers or mates) share foundations, or differ in their regulation. Ultimately these studies should help us understand how selective relationships of different types evolve.